# Serum Adipokines, Growth Factors, and Cytokines Are Independently Associated with Stunting in Bangladeshi Children

**DOI:** 10.3390/nu11081827

**Published:** 2019-08-07

**Authors:** Muttaquina Hossain, Baitun Nahar, Md. Ahshanul Haque, Dinesh Mondal, Mustafa Mahfuz, Nurun Nahar Naila, Md. Amran Gazi, Md. Mehedi Hasan, Nur Muhammad Shahedul Haque, Rashidul Haque, Michael B Arndt, Judd L Walson, Tahmeed Ahmed

**Affiliations:** 1Nutrition and Clinical Services Division, icddr,b, Dhaka 1212, Bangladesh; 2Enteric and Respiratory Infections, icddr,b, Dhaka 1212, Bangladesh; 3PATH, Seattle, WA 98109, USA; 4Department of Global Health, University of Washington, Seattle, WA 98109, USA; 5Department of Pediatrics, University of Washington, Seattle, WA 98109, USA; 6Childhood Acute Illness and Nutrition Network, Nairobi 00200, Kenya; 7Departments of Medicine, University of Washington, Seattle, WA 98109, USA; 8Department of Epidemiology, University of Washington, Seattle, WA 98109, USA; 9James P. Grant School of Public Health, BRAC University, Dhaka 1212, Bangladesh

**Keywords:** adipokines, peptides, stunting, children, Bangladesh

## Abstract

Growth in young children is controlled through the release of several hormonal signals, which are affected by diet, infection, and other exposures. Stunting is clearly a growth disorder, yet limited evidence exists documenting the association of different growth biomarkers with child stunting. This study explored the association between different growth biomarkers and stunting in Bangladeshi children. A quasi-experimental study was conducted among 50 stunted (length-for-age *Z*-score (LAZ) < −2 SD) and 50 control (LAZ ≥ −2 SD) children, aged 12–18 months, residing in a Bangladeshi slum. The enrolled stunted children received an intervention package, which included food supplementation for three months, psychosocial stimulation for six months, and routine clinical care on community nutrition center at the study field site. The controls received routine clinical care only. All children were clinically screened over the study period. Length, weight, fasting blood and fecal biomarkers were measured. All biomarkers levels were similar in both groups except for oxyntomodulin at enrolment. Leptin (adjusted odds ratio, AOR: 4.0, *p* < 0.01), leptin–adiponectin ratio (AOR 5.07 × 10^8^, *p* < 0.01), insulin-like growth factor-1 (IGF-1) (AOR 1.02, *p* < 0.05), and gamma interferon (IFN-γ) (AOR 0.92, *p* < 0.05) levels were independently associated with stunting at enrolment. Serum leptin, leptin–adiponectin ratio, interleukin-6 (IL-6), IL-10, tumor necrosis factor-alpha (TNF-α), and fecal alpha-1-antitrypsin (AAT) levels increased significantly (*p* < 0.001), while IFN-γ levels significantly decreased among stunted children after six months of intervention. Leptin, leptin–adiponectin ratio, IGF-1, and IFN-γ are independently associated with stunting in Bangladeshi children. This trial was registered at clinicaltrials.gov as NCT02839148.

## 1. Introduction

Stunting (length-for-age *Z*-score (LAZ) < −2 SD) or linear growth faltering has long-term devastating effects on child health. Stunting is associated with increased childhood morbidity and mortality, as well as reduced developmental and decreased cognitive function, and is also associated with elevated risk for developing chronic diseases in adulthood [1]. Globally, 151 million children under the age of five are stunted, and more than half of these live in southern Asia [2]. Bangladesh is a lower-middle-income country in southern Asia where one in three children under five are stunted [3]. The burden is particularly high among 12–24-month-old children living in slums. Studies conducted among Bangladeshi slum children identified child age, sex, poor length at birth, maternal undernutrition, low maternal educational attainment, consumption of untreated drinking water, and poor socioeconomic status as potential risk factors for stunting [4,5]. Despite these known biological and socio-economic risk factors, there are several less explored biological aspects in the development of stunting. One such less explored aspect is the relationship of several gut and fat cell-derived hormones and peptides with child linear growth. 

Growth is a dynamic process and influenced by nutritional, hormonal, and neuronal factors. The linear growth during infancy is principally dependent on nutrition, although endocrine factors like the growth hormone/insulin-like growth factor-1 (GH/IGF-1) axis play an increasingly important role from one year of age [6]. The GH/IGF-1 axis is an important regulator of linear growth through bone formation and resorption [7]. In addition, adipocytes play an important role in child growth through energy regulation and glucose homeostasis [6]. The GH/IGF-1 axis acts to control various metabolic functions through the secretion of adipokines. Adiponectin and leptin are considered the most important adipokines [8]. Adiponectin plays a role in the metabolism of carbohydrates and fatty acids, insulin resistance, inflammation, and bone metabolism. Studies showed that serum levels of adiponectin inversely correlate with body mass index (BMI), fat mass, and bone mineral density in children and adults [9]. Leptin stimulates linear growth by affecting the proliferation, hypertrophy, and calcification of the growth-plate cartilage [10] through the activation of fibroblast growth factors (FGFs) [11]. In addition, leptin is also known as a satiety factor, which regulates body weight by suppression of appetite and stimulation of energy expenditure [12]. Other appetite-suppressing hormones include glucagon-like peptides (GLP-1 and GLP-2) and oxyntomodulin (OXM); in contrast, ghrelin is one of the hormones that stimulate appetite [13]. However, this appetite pathway is disrupted during infection and inflammation, conditions that are particularly common in children living in low-resource settings [14].

Immune dysfunction is directly related to the pathological processes in malnutrition, including malabsorption, increased metabolic demand, dysregulation of the growth hormone and hypothalamic–pituitary–adrenal axis, and greater susceptibility to infection [15]. Interferon-gamma (IFN-γ), tumor necrosis factor-alpha (TNF-α), and interleukins (e.g., IL-6, IL-10) are cell-signaling cytokines that activate and drive differentiation of immune cells upon infection [16]. During infection, elevated levels of cytokines like TNF-α lead to increased blood leptin concentrations, diminished appetite [17], and antagonized growth [18,19]. Recurrent enteric infections and chronic mucosal inflammation in children exposed to poor sanitation and hygiene lead to a condition named environmental enteric dysfunction (EED) [20]. EED is considered as one of the key determinants of linear growth failure in early years of life [21]. The possible mechanisms contributing to growth failure in EED include intestinal leakiness and increased gut permeability, gut inflammation, systemic inflammation, and nutrient malabsorption [20]. Some candidate biomarkers that are indicative of gut inflammation and increased intestinal permeability caused by EED in children include fecal myeloperoxidase (MPO), neopterin (NEO), and alpha-1-antitrypsin (AAT) [20]. Raised fecal MPO and NEO levels indicate gut inflammation, and elevated fecal AAT indicates increased intestinal permeability and protein loss [22]. 

Little is known about the complex interactions and relationships among the gut hormones, human growth factors, EED markers, cytokines, and inflammatory markers with child linear growth, especially during the first two years of life. Although studies demonstrated strong correlations between certain hormones and fetal growth [23,24], the role of these hormones on linear growth after one year of birth is less well understood. The typical circulating levels of these biomarkers levels in Bangladeshi children under the age of two are not known. Therefore, we aimed to explore the serum levels of circulating gut hormones and adipokines (ghrelin, OXM, GLP-1, leptin, and adiponectin), human growth factors (FGF-19, FGF-21, IGF-1, and GLP-2), fecal markers of EED, cytokines (IL-1β, IL-4, IL-6, IL-10, TNF-α, and IFN-γ), and systemic inflammation markers (C-reactive protein) among both stunted and control children. We also explored the association between different biomarkers and stunting. The comparison of different biomarkers levels between stunted and control children at enrolment were also examined. In addition, this paper reports the association of child length and BMI changes, with serum adipokines and peptides at six months post-enrolment among stunted and control children. This result would help to inform the government, policy-makers, donors, and academia about the different biomarkers as a useful proxy indicator of growth failure or recovery, which could be measured in future research or in intervention programs designed to reduce child stunting. The measurement of different biomarkers, especially adipokines, growth factors, and cytokines in stunted children, would further help in implication in the prevention of stunting in young children and also to find a suitable solution to treat growth disorders in young children.

## 2. Materials and Methods 

### 2.1. Sample Population and Study Design

A quasi-experimental study was conducted on a convenience sample size (*n* = 100) of 50 stunted (length-for-age *Z*-score, LAZ < −2 SD) and 50 control children (LAZ ≥ −2 SD) of both sexes aged between 12 and 18 months, residing in the Mirpur slum of Dhaka city, Bangladesh. Due to ethical reasons, we could not include a “stunted, no intervention” group, and the study also lacked a “non-stunted intervention” group due to resource constraints, which makes this study design weaker than an ideal intervention trial. The control children were selected and enrolled after matching for age and sex with the stunted children. The control-to-case ratio was 1:1. The participants included in the cohorts and the analysis workflow are described in Figure 1, based on the Consolidated Standards of Reporting Trials diagram [25] and the Strengthening the Reporting of Observational Studies in Epidemiology (STROBE) guideline [26]. The details of the study site were published elsewhere [27]. The parents/caregivers who provided consent for their child to participate in the study were included in the study and followed up for the next six months. Children were excluded if they had any acute illness or known chronic disease such as tuberculosis, or any congenital anomalies such as trisomy-21, cleft lip or palate. The intervention and control children were selected from different blocks in the same community, leaving a “buffer zone” to minimize the risk of contamination. 

### 2.2. Intervention Description

The enrolled 50 stunted children received an intervention package, which included food supplementation (FS) for three months, psychosocial stimulation (PS) for six months [28], and routine clinical care on community nutrition center (CNC) at the study field site. The controls received routine clinical care only but no FS or PS. The food supplementation included one boiled hen’s egg (55 g approximately) and 150 mL of ultra-high temperature treated cow’s milk. The PS comprised play sessions and parental counseling [28] on improving child-rearing practices and enhancing mother–child interaction for optimal child growth and development. The sessions were conducted weekly for the first month, fortnightly for second and third months, and then monthly for the next three months. A study physician provided routine clinical care to all enrolled children, which included physical examination, micronutrient powder supplementation, growth monitoring, treatment of intercurrent illness(es), health and nutrition education (counseling on infant and young child feeding practices, personal hygiene and hand washing), immunization under the Expanded Program of Immunization schedule, and de-worming medication if not given over the last six months from enrolment.

### 2.3. Data Collection

Trained field staff interviewed the parents or caregivers to collect socioeconomic and household information at enrolment by using a structured and pre-tested questionnaire. They also assessed the intake and leftovers of offered food (one egg and 150 mL of milk) by stunted children using a standard format. Stool and blood samples were collected at enrolment and at six months post-enrolment. Stool samples were collected without fixative and kept frozen at −80 °C pending processing. Fasting serum was obtained via centrifugation of the blood. All samples from each individual were collected within a seven-day window.

### 2.4. Anthropometric Measurements

Trained field staff measured every child’s body weight using Seca scales with an accuracy of 10 g, and measured length using a Seca length board with 1 mm accuracy at community nutrition center (CNC). The measurements were taken once every month for six months, at a fixed time in the morning with minimal clothing and without any shoes. Mothers’ weight, height, and BMI were also measured once on enrolment using the standard procedure.

### 2.5. Laboratory Analyses

Firstly, 5 mL of venous blood was collected from children after overnight fasting into EDTA tubes, centrifuged for 10 min at 4000 rpm, and immediately aliquoted. Aliquots were stored at −80 °C until analysis. All the biomarkers were analyzed using commercially available enzyme-linked immunosorbent assay (ELISA) kits according to the manufacturers’ instructions. Inflammatory markers such as C-reactive protein (CRP), tumor necrosis factor (TNF)-α, and gamma-interferon (IFN-γ), cytokines such as interleukins (ILs) 1β, 4, 6, 10, and 12, adipokines such as leptin and adiponectin, and growth factors such as fibroblast growth factor (FGF)-19, 21 and insulin-like growth factor (IGF)-1 were measured using ELISA kits from R&D Systems Inc, (Minneapolis, MN 55413, USA). Acyl-ghrelin, oxyntomodulin, and GLP-1 were measured using ELISA kits from Lifespan Biosciences (Fourth Avenue, Seattle, WA, USA), and GLP-2 using an ELISA kit from Merck Millipore (Burlington, MA, USA). 

Stool samples were analyzed for myeloperoxidase (MPO) (Alpco, Salem, NH, USA), neopterin (NEO) (GenWay Biotech, San Diego, CA, USA), and alpha-1 antitrypsin (AAT) (Biovendor, Chandler, NC, USA) using ELISA. Final absorbances were taken at 450 nm with a correction absorbance at 630 nm using an ELx 808 ELISA plate reader. Concentrations were calculated against standards for each of the biomarkers. Samples were pre-treated according to the kit manuals provided by the manufacturers, and the final dilution of serum and fecal biomarkers was determined by selecting the most appropriate concentration of a biomarker falling in the linear range of the standard curve. All laboratory analyses were performed by trained laboratory personnel at icddr,b laboratories.

### 2.6. Statistical Analysis

A trained data manager using Microsoft Access 2007 performed data entry and validation of questionnaires and anthropometry forms. Statistical analyses were performed using STATA version 13.0. Normality of data distribution was tested using the Shapiro–Wilk test. To summarize the data, proportion estimates were used for categorical variables, means with standard deviation were used for normally distributed quantitative variables, and median estimates with interquartile ranges (IQRs) were used for asymmetric quantitative variables. Counts with a normal distribution were compared by Student’s *t*-test, and variables with asymmetric distribution were compared by Wilcoxon rank-sum/Mann–Whitney U test. Qualitative variables were compared by the χ^2^ distribution test. The association of different biomarkers with stunting at enrolment was explored using a logistic regression model. Individual laboratory parameters were the predictor, and child-stunting status at enrolment was the outcome. The model was adjusted for potential confounders. Potential confounders related to the exposure and/or the outcomes of interest in the bivariate models with a *p*-value <0.25 were included in the multivariable models. Models with the lowest Akaike information criterion (AIC) and Bayesian information criterion (BIC) were considered as final models. Estimated associations were described as odds ratios (ORs) with 95% confidence intervals (CIs). Since the distribution of the majority of measured biomarker values deviated from a normal distribution, non-parametric tests were performed. The non-parametric Wilcoxon test for paired samples was used to compare different laboratory parameters at different times among the stunted children. The Mann–Whitney U test for unpaired samples was applied to compare different laboratory parameters between the stunted and control group at enrolment. An intention-to-treat analysis was applied for longitudinal analysis, and missing values were considered as missing at random. Multivariable linear regressions were done to find out the association of different adipokines, peptides, and growth factors on the length and BMI change among the groups after controlling for child age, sex, FS and PS for stunted, and age and sex for the control group. To evaluate potential associations between biomarkers, Spearman’s rank correlations were calculated. A *p*-value <0.05 was considered significant. 

### 2.7. Ethical Considerations 

Ethical approval for this study was obtained from the Institutional Review Board (IRB) of icddr,b (protocol no; PR-16005; version 2; 3 March 2016). The ethics committees approved the consent format prior to data collection. Written, informed consent of the parents/guardians of the children was taken before enrolment into the study, and confidentiality of all personal information was protected. None other than the study personnel had access to the information on personal identification and other sensitive information. All information collected from the participants including the results of laboratory tests were locked in a secure place under the responsibility of the principal investigator of the study and were coded in such a way that no one would be able to trace the participants. This trial was registered at clinicaltrials.gov as NCT02839148.

## 3. Results

### 3.1. Basic Socio-Demographic Characteristics

One hundred children were enrolled in the study; among them, 50 were stunted and 50 were healthy controls (Table 1). Thirty-nine stunted children and 24 control children out of 100 enrolled children completed six months follow-up. The loss to follow-up was mostly due to the loss of contact with caregivers, migration from the study area, and refusal to provide a biological sample. Data of all children at both time points were analyzed. The mean age of the children at enrolment was 14 months. About half of the enrolled children were female, and half were anemic in each group. The controls had significantly better weight, mid-upper arm circumference (MUAC), and body mass index (BMI) compared to the stunted children (*p* < 0.001). The stunted children were significantly more underweight (weight-for-age *Z*-score −2.09 vs. −1.03) and wasted (weight-for-length *Z*-score −1.2 vs. 0.66) compared to controls (*p* < 0.001 respectively). Mothers’ age, level of education, and occupation were comparable across the groups (Table 1). 

### 3.2. Differences in Biomarker Levels at Enrolment between Stunted and Control Children

The results of the Mann–Whitney U test show that oxyntomodulin levels were significantly lower at enrolment in control children compared to their stunted peers (*p* < 0.05 and *p* < 0.001). Leptin and leptin–adiponectin ratio levels were lower in stunted children compared to controls at enrolment although not statistically significant (Table 2).

### 3.3. Biomarkers Associated with Stunting

Table 3 shows the results of logistic regression analysis. In bivariate analysis, there were no significant associations between gut hormones, adipokines, human growth factors, EED markers, cytokines, or systemic inflammatory markers with child stunting status. Fasting serum leptin (adjusted odds ratio (AOR) = 3.9; 95% confidence interval (CI): 1.3–10.9), leptin–adiponectin ratio (AOR = 4.31 × 10^8^; 95% CI: 696–2.67 × 10^14^), insulin-like growth factor-1 (IGF-1) (AOR = 1.02; 95% CI: 1.0–1.04), and gamma interferon (IFN-γ) (AOR = 0.92; 95% CI: 0.85–0.98) were found independently significantly associated with stunting, after adjusting for child age, sex, weight-for-age *Z*-score, and hemoglobin status. Serum levels of leptin, leptin–adiponectin ratio, and IGF-1 were significantly elevated among stunted children at enrolment. None of the EED and systemic inflammatory markers were significantly associated with child stunting at enrolment. 

### 3.4. Changes in Biomarker Levels among Stunted Children over Time 

Among the stunted children, fasting serum ghrelin, myeloperoxidase, IL-1B, IL-4, IL-12, and IFN-γ levels declined significantly (*p* < 0.05 and *p* < 0.001) between enrolment and month six of follow-up (Table 4). Serum leptin, leptin–adiponectin ratio, IL-6, IL-10, TNF-α, and fecal AAT levels increased significantly (*p* < 0.001) from enrolment, after six months of food supplementation and psychosocial intervention. 

### 3.5. Association of Child Length and BMI Changes with Serum Adipokines and Peptides at Six Months Post-Enrolment among the Stunted and Control Children

To verify the relationship between changes in length and BMI with those of adiponectin, leptin, leptin–adiponectin ratio, IGF-1, IFN-γ, and IL-6 among stunted and control children at six months post-enrolment; we used linear regression models (Table 5). We did not observe any significant association between child length gain and BMI change with leptin, adiponectin, leptin–adiponectin ratio, IGF-1, IFN-γ, and IL-6 levels in stunted children after controlling for food intake and child psychosocial development. Statistical adjustment for child age and sex did not alter these findings. However, we observed a significant negative association between child BMI changes and adiponectin among control children, after adjustment for child age and sex.

### 3.6. Correlation among the Biomarkers

At enrolment, GLP-1, IL-4, and IL-6 were positively correlated with IL-12 (ρ = 0.53, 0.59, and 0.52, *p* < 0.05), and TNF was positively correlated with IL-10 (ρ = 0.61, *p* < 0.05) among stunted children. CRP was positively correlated with IL-6 at six months post-enrolment (ρ = 0.58, *p* < 0.05) among stunted children. Leptin was positively correlated with oxyntomodulin both at enrolment and six months post-enrolment in control children (ρ = 0.82, *p* < 0.05). None of the gut hormones and adipokines were significantly correlated with each other at six months among the stunted children and control children, as shown in Appendix A). 

## 4. Discussion

This study evaluated levels of gut hormones, adipokines, human growth factors, fecal markers of EED, cytokines, and systemic inflammation marker levels in a cohort of Bangladeshi stunted and control children at enrolment and six months post-enrolment. To our knowledge, most prior studies on gut hormones, adipokines, human growth factors, and cytokines were conducted in acutely ill populations [29,30], pregnant women [31], neonates [32,33], and pre-pubertal children [34,35]. This study reports circulating levels of these biomarkers among Bangladeshi children less than two years of age. The study also revealed that fasting serum leptin, leptin–adiponectin ratio, IGF-1, and IFN-γ are independently associated with stunting in Bangladeshi children under two years of age. 

We found lower levels of leptin among stunted children compared to controls at enrolment before the intervention. During prolonged nutritional deprivation, the decreased energy intake diminished fat mass, and declining insulin concentrations suppressed leptin production [36]. Low leptin levels stimulate the hypothalamic–pituitary–adrenal axis and possibly the hypothalamic–pituitary–GH axis to maintain the high cortisol and GH levels necessary for effective lipolysis to ensure a fuel (fatty acids) supply for the metabolism of the brain and peripheral tissue during nutritional deprivation. By this process, leptin ensures substrate diversion away from linear growth toward metabolic homeostasis [36]. This situation improves as the child grows up. At the end of six months of intervention, higher levels of leptin and leptin–adiponectin ratio were observed among the stunted children [37]. We firstly speculated this alteration in adipokines in stunted children may be due to fat mass increase [38] as a result of the intervention (food supplementation and psychosocial stimulation). However, we could not rule out the effect of fat mass on hormonal changes, as it was not measured in the current study. In addition, there was no significant association found between child length and BMI changes with any of the serum adipokine and peptide levels among stunted children with food supplementation and psychosocial stimulation.

Leptin production occurs after increases in insulin in response to feeding, and a decrease in leptin concentrations follows decreases in insulin during fasting [39]. Both leptin and IGF-1 were independently associated with stunting in this study. There was a poor correlation between IGF-1 and leptin in stunted children at both enrolment and six months post-enrolment. These results suggest that the stimulatory effect of leptin on growth under chronic caloric restriction is not dependent on circulating IGF-1 [40]. The length gain over time among both stunted and control children was slightly higher among stunted children although not statistically significant. Thus, leptin’s role as a skeletal growth factor may be GH/IGF-1-independent [40,41]. 

Sub-optimal nutrition and chronic inflammation mediated by pro-inflammatory cytokines contribute to the underlying etiology of growth retardation [18]. Levels of CRP and most of the pro-inflammatory cytokines (IL-6, IL-12, and TNF-α,) were lower among stunted children than controls at baseline. A similar finding was shown by an Egyptian study on children with protein-energy malnutrition [42]. The pro-inflammatory immune response, mediated by the low leptin levels, is also impaired during malnutrition [39]. We also found a lower level of leptin among stunted children at enrolment compared to controls. One possible explanation for this could be the alteration of immune responses among stunted children. During chronic malnutrition, there is a reduction in bone marrow, resulting in a decrease in the production of IL-6 and TNF-α by bone marrow cells [43]. In addition, malnourished children are more susceptible to infections. Therefore, the body depresses cellular immunity as an adaptive response to prevent autoimmune reactions [15]. Moreover, undernutrition weakens the body’s defenses against infections, which in turn trigger the inflammation along with markers of inflammation; thus, the vicious cycle of infection malnutrition continues to deteriorate the child’s condition [15]. The picture would be incomplete without mentioning the wasting effect of the infection itself [44]. The stunted children in our study were comparatively more wasted and underweight than the comparison group, which could be another explanation of reduced cytokine production. Several studies reported marked deficits in IL-6, TNF-α, IL-1, and IFN-γ levels in malnourished children [45,46,47]. 

IL-6 is considered the most important inducer of hepatocyte synthesis of acute-phase proteins in response to infection/inflammation stimuli, and CRP is produced mainly in the liver in response to IL-6 [48]. A previous study conducted in our current study area confirmed the presence of more infection among malnourished children than their peers [49]. Our study children were not an exception, and the presence of raised IL-6 level indicates the presence of underlying infection. However, the IL-6 levels increased in stunted children after six months of intervention, not at baseline, which contradicts the findings from other studies [49,50]. We also observed a moderate positive correlation between IL-6 and CRP levels in stunted children after six months of intervention, which is explainable. Despite constant exposure to the infectious stimulus, the lack of cell-mediated immune response in stunted children might have limited the expression of inflammatory cells to the stimuli at baseline [39]. The raised levels level of IL-6 in stunted children after six months of intervention might be due to the improvement in cell-mediated immune response toward the infection/inflammation stimuli, because of essential micro and macronutrients rich food supplementation [51,52]. However, these findings need to be further explored in future studies. There might be a concern of cow’s milk and hen’s egg allergy regarding the increase in IL-6 levels, but none of our stunted study children showed or reported any sign of food allergy throughout the intervention period.

Studies showed stronger support for the link between intestinal inflammation and systemic inflammation, and between intestinal inflammation and stunting [53]. However, none of the EED markers were significantly associated with stunting at enrolment in our study. A recent study conducted among rural Bangladeshi children also found similar results [54]. In addition, EED markers were not significantly different between the control and stunted group. This could be explained by the transient nature of EED markers in young children [19,54], and the declining trend of EED biomarker levels after 12 months of age [19]. EED assessed at enrolment is indicative of present status rather than prior or chronic illness and, thus, had no association with child stunting status, which is a chronic form of malnutrition. 

### Strengths and Limitations

The presence of a community nutrition center at a feasible distance from all participants and dedicated trained staff ensured the collection of blood samples at fasting during a fixed time, which helped to reduce the effect of diurnal variations in hormones on measurements. In addition, all the biomarkers were tested in a single batch to reduce potential batch effects and inter-tester variation, which strengthened the validity of the study findings.

This study was limited by its relatively small sample size. In addition, there were fewer available blood samples from control children during follow-up as caregivers opposed repeated venipuncture in otherwise “healthy” children. In addition, the control children did not receive any intervention; thus, their caregivers were reluctant to provide information and biological samples. Also, they were not frequently contacted by the research team, in order to avoid the “Hawthorne effect”; as a result, drop-out rates were relatively higher among the controls. The study results might be biased due to this higher drop-out rate among controls. Although all children were fasting at enrolment, diet is known to affect levels of many hormonal markers, particularly leptin. Therefore, differences in dietary patterns between the included groups may have resulted in some of the observed marker differences. Several other co-factors of stunting such as common co-morbidities (i.e., fever, diarrhea, pneumonia, etc.), food insecurity, micronutrient deficiency, body composition estimation, or antibiotic intake were not assessed due to time and resource limitation. Due to the high prevalence of stunting among slum children aged between 12 and 18 months, all of the control children were considered at risk of stunting and may not reflect what would be observed in truly “healthy” controls. In addition, we could not include “stunted, no intervention” group due to ethical reasons, which might have created a spurious effect on the findings. Hence, the findings needed to be interpreted cautiously. Due to the lack of a true control group, this study could not confirm that the specific effect of the observed changes in the biomarkers occurred as a result of the intervention, as opposed to a naturally occurring effect based on the aging process of the children. These findings should be confirmed in subsequent studies that ideally would allow for some causal inference as to the role of these biomarkers in the development of stunting. Finally, these findings may not be generalizable to settings other than a slum, although many of the biomarkers identified herein were associated with malnutrition in other studies.

## 5. Conclusions

This study found that fasting serum leptin, leptin–adiponectin ratio, IGF-1, and IFN-γ were independently associated with stunting in Bangladeshi children under the age of two. These markers may be useful proxy indicators of growth failure or recovery that could be measured in future research or in intervention programs designed to reduce stunting. To improve generalizability and for documenting causal relationships between these adipokines, human growth factors, and cytokines and growth in young children, longitudinal research with larger sample sizes is needed. 

## Figures and Tables

**Figure 1 nutrients-11-01827-f001:**
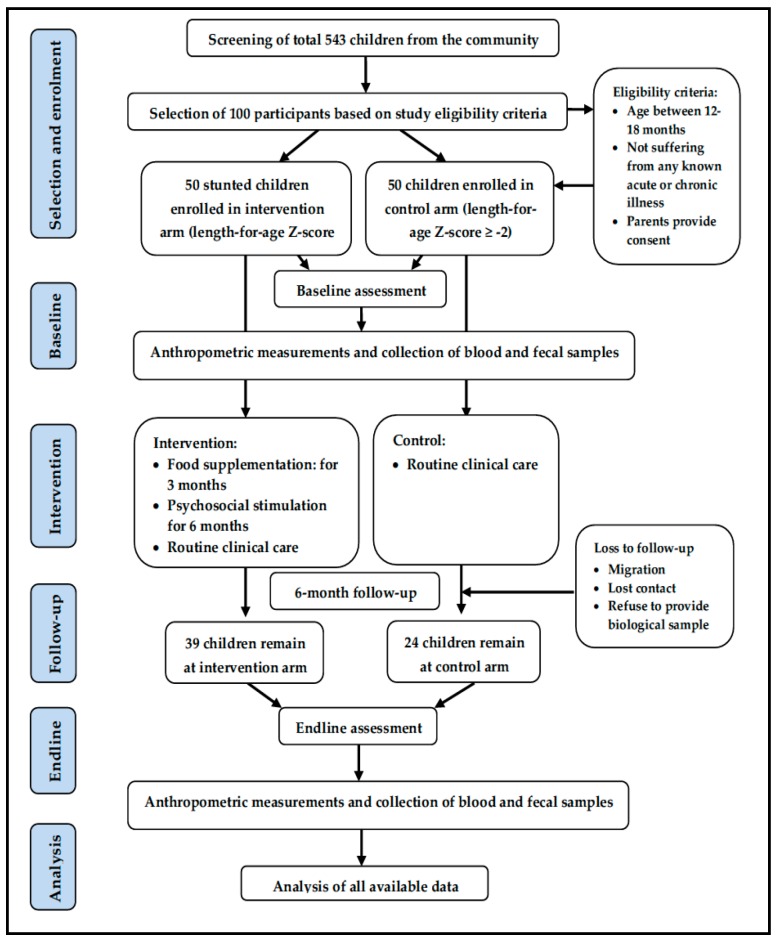
Cohorts, participants, and analyses flow chart.

**Table 1 nutrients-11-01827-t001:** Socio-demographic parameters among stunted and control children at enrolment.

Variables	Stunted Children (*n* = 50)	Control Children (*n* = 50)	*p*
Child characteristics			
Age, months	14.4 ± 2.01	14.6 ± 2.12	0.721
Female, %	52.6	47.4	0.545
Weight, kg	7.6 ± 7.4	8.7 ± 8.5	<0.001
Length, cm	71.1 ± 2.4	74.7 ± 2.8)	<0.001
MUAC	13.2 ± 0.86	14.1 ± 0.95	<0.001
Weight-for-age *Z*-score	−2.09 ± 0.59	−1.03 ± 0.68	<0.001
Length-for-age *Z*-score	−2.43 ± 0.26	−1.18 ± 0.39	<0.001
Weight-for-length *Z*-score	−1.2 ± 0.76	−0.66 ± 0.83	<0.001
BMI-for-age *Z*-score	−0.85 ± 0.80	−0.47 ± 0.84	0.021
Hemoglobin, gm/dL	9.8 ± 1.5	9.5 ± 1.3	0.258
Anemia, %	54.1	45.8	0.267
Exclusive breast feeding, %	45.4	54.5	0.205
Maternal characteristics			
Age, years	23.8 ± 5.3	24.6 ± 5.5	0.459
Education level, %			0.909
None	20	28	
Up to higher secondary school	40	34	
Graduate	40	38	
Occupation, %			0.509
Housewife	86	92	
Readymade garments worker, school teacher, small business	14	8	

Abbreviations: MUAC—mid-upper arm circumference; BMI—body mass index. Continuous variables are presented by means ± SD and categorical variables are presented as the percentage of participants (%).

**Table 2 nutrients-11-01827-t002:** The difference in different biomarkers levels at enrolment between stunted and control children.

Variables	At Enrolment		
	Stunted Children (*n* = 50)	Control Children (*n* = 50)	*p*
Gut hormones and adipokines			
Ghrelin, pg/mL	35.7 (24.4–70.6)	27.5 (19.2–40.6)	0.07
Leptin, µg/L	0.95 (0.57–1.4)	1.19 (0.74–1.78)	0.22
Adiponectin, µg/mL	14.5 (9.7–17.2)	13.5 (9.1–19.4)	0.85
OXM, ng/mL	60.8 (40.7–70.3)	42.3 (32.8–61.8)	0.02
GLP-1, pg/mL	133.8 (91.2–245.1)	128.7 (95.2–230.3)	0.94
Leptin–adiponectin ratio	0.06 (0.04–0.11)	0.08 (0.04–0.12)	0.49
Human growth factors			
GLP-2, ng/mL	3.84 (2.1–5.12)	3.56 (2.2–4.2)	0.24
IGF-1, ng/mL	48.9 (36.1–83.2)	42.3 (23.8–71.3)	0.20
FGF-19, pg/mL	98.5 (66.5–162.8)	117.9 (69–255.5)	0.22
FGF-21, pg/mL	225.6 (113.2–392)	169.7 (85.1–325.9)	0.32
Environmental enteric dysfunction markers			
AAT, mg/gm	0.50 (0.26–0.93)	0.29 (0.17–0.95)	0.44
MPO, ng/gm	1932 (756.5–5238.5)	2064.5 (858–3560)	0.70
NEO, nmol/L	5944 (3423–8323)	4867.5 (3950–5692)	0.13
Cytokines			
IL-1B, pg/mL	0.84 (0.68–0.99)	0.83 (0.68, 1.15)	0.82
IL-4, pg/mL	3.41 (2.0–3.8)	2.07 (2.07–3.41)	0.34
IL-6, pg/mL	0 (0–1.2)	0 (0–3.2)	0.81
IL-10, pg/mL	17.3 (12.8–27.6)	15.3 (10–20.8)	0.07
IL-12, pg/mL	1.05 (0.08–2.4)	0.57 (0.28–1.4)	0.26
TNF, pg/mL	1.49 (0–7.7)	2.6 (0–5.5)	0.84
IFN-γ, pg/mL	0 (0–2.5)	0 (0–0)	0.82
Systemic inflammatory marker			
CRP, mg/L	1.18 (0.58, 2.2)	1.54 (0.41, 3.8)	0.47

Abbreviations: AAT: alpha-1 antitrypsin; CRP: C-reactive protein; FGF: fibroblast growth factor; IFN-γ: gamma interferon; GLP: glucagon-like peptide; ILs: interleukins; IGF: insulin-like growth factor; MPO: myeloperoxidase; NEO: neopterin; OXM: oxyntomodulin; TNF-α: tumor necrosis factor-alpha. All values are medians; 25th–75th percentiles are shown in parentheses; *p*-values were determined using the Mann–Whitney U test.

**Table 3 nutrients-11-01827-t003:** Association of different biomarkers with child stunting at enrolment.

Variables	Stunted (*n* = 50)	Control (*n* = 50)	Crude Model		Adjusted Model ^1^	
			OR (95% CI)	*p*	OR (95% CI)	*p*
Gut hormones and adipokines						
Ghrelin, pg/mL	50.5 ± 39.2	37.9 ± 29.2	1.0 (0.99, 1.02)	0.08	1.0 (0.98, 1.02)	0.469
Leptin, µg/L	1.23 ± 1.23	1.33 ± 0.88	0.92 (0.63, 1.3)	0.665	4.0 (1.4, 11.3)	0.009
Adiponectin, µg/mL	14.2 ± 5.2	14.5 ± 6.2	0.98 (0.92, 1.06)	0.75	0.97 (0.87, 1.08	0.655
OXM, ng/mL	58.5 ± 20.9	51.6 ± 28.0	1.01 (0.99, 1.02)	0.18	1.01 (0.99, 1.04)	0.149
GLP-1, pg/mL	286.7 ± 436.7	235.2 ± 293.7	1.0 (0.99, 1.0)	0.49	1.0 (0.99, 1.0)	0.893
Leptin–adiponectin ratio	0.16 ± 0.41	0.08 ± 0.06	0.07 (0.0003, 18.58)	0.35	5.07 × 10^8^ (813.04, 3.16 × 10^14^)	0.003
Human growth factors						
GLP-2, ng/mL	3.7 ± 2.3	3.3 ± 1.9	1.1 (0.92, 1.33)	0.28	0.99 (0.74, 1.3)	0.947
IGF-1, ng/mL	58.6 ± 34.6	51.6 ± 35.1	1.0 (0.99, 1.01)	0.32	1.02 (1.0, 1.04)	0.016
FGF-19, pg/mL	135.4 ± 119.1	175.9 ± 154.6	0.99 (0.99, 1.0)	0.15	0.99 (0.99, 1.0)	0.116
FGF-21, pg/mL	359.3 ± 467.5	249.5 ± 233.2	1.0 (0.99, 1.0)	0.16	1.0 (0.99, 1.0)	0.780
Environmental enteric dysfunction biomarker						
AAT, mg/gm	1.3 ± 2.1	1.6 ± 2.7	0.94 (0.80, 1.11)	0.53	1.01 (0.80, 1.29)	0.874
MPO, ng/gm	5254.4 ± 7646.7	4112.1 ± 5783.9	1.0 (0.99, 1.0)	0.40	1.0 (0.99, 1.0)	0.759
NEO, nmol/L	5944 ± 3423	5081.6 ± 2228.9	1.0 (0.99, 1.0)	0.22	1.0 (0.99, 1.0)	0.229
Cytokines						
IL-1b, pg/mL	0.97 ± 0.63	0.95 ± 0.47	1.06 (0.52, 2.16)	0.86	1.05 (0.38, 2.9)	0.911
IL-4, pg/mL	3.4 ± 2.9	3.02 ± 1.6	1.07 (0.89, 1.28)	0.43	0.98 (0.77, 1.25)	0.897
IL-6, pg/mL	1.2 ± 2.8	4.9 ± 11.3	0.90 (0.82, 0.99)	0.04 *	0.89 (0.75, 1.05)	0.185
IL-10, pg/mL	19.5 ± 8.7	17.2 ± 10.5	1.02 (0.98, 1.07)	0.23	0.95 (0.88, 1.0)	0.092
IL-12, PG/mL	5.9 ± 19.2	1.2 ± 2.2	1.09 (0.93, 1.27)	0.27	1.06 (0.89, 1.3)	0.462
TNF-α, pg/mL	4.4 ± 5.4	3.9 ± 4.7	1.01 (0.94, 1.1)	0.63	0.95 (0.83, 1.09)	0.507
IFN-γ, pg/mL	3.3 ± 6.6	5.6 ± 12.4	0.97 (0.93, 1.01)	0.25	0.92 (0.85, 0.98)	0.023
Systemic inflammatory marker						
CRP, mg/L	1.8 ± 1.8	2.7 ± 2.9	0.86 (0.73, 1.02)	0.09	0.74 (0.53, 1.02)	0.07

Abbreviations: AAT: alpha-1 antitrypsin; CI: confidence interval; CRP: C-reactive protein; FGF: fibroblast growth factor; IFN-γ: gamma interferon; GLP: glucagon-like peptide; ILs: interleukins; IGF: insulin-like growth factor; MPO: myeloperoxidase; NEO: neopterin; OR: odds ratio; OXM: oxyntomodulin; TNF-α: tumor necrosis factor-alpha. All values are means ± SD, and *p*-values were determined by the logistic regression model. ^1^ Logistic regression model was adjusted for child age, child sex, exclusive breastfeeding, weight-for-age *Z*-score, and hemoglobin at enrolment. * *p* < 0.05.

**Table 4 nutrients-11-01827-t004:** Changes in different biomarkers among stunted children over time.

Variables	Stunted Children		
	At Enrolment (*n* = 50)	At 6 Months (*n* = 39)	*p*
Gut hormones and adipokines			
Ghrelin, pg/mL	35.7 (24.4–70.6)	27.3 (11.9–43.2)	0.013
Leptin, µg/L	0.95 (0.57–1.4)	1.3 (0.90–1.89)	0.005
Adiponectin, µg/mL	14.5 (9.7–17.2)	14.2 (10.7–18.1)	0.379
OXM, ng/mL	60.8 (40.7–70.3)	57.1 (47.7–62.4)	0.911
GLP-1, pg/mL	133.8 (91.2–245.1)	141 (107–210)	0.364
Leptin–adiponectin ratio	0.06 (0.04–0.11)	0.11 (0.07–0.15)	0.003
Human growth factors			
GLP-2, ng/mL	3.84 (2.1–5.12)	3.28 (2.30–3.92)	0.053
IGF-1, ng/mL	48.9 (36.1–83.2)	46.7 (41.7–59.2)	0.503
FGF-19, pg/mL	98.5 (66.5–162.8)	99.9 (70.4–171.9)	0.665
FGF-21, pg/mL	225.6 (113.2–392)	171.1 (122.6-270.6)	0.175
Environmental enteric dysfunction markers			
AAT, mg/gm	0.50 (0.26–0.93)	3.54 (0.26–6.03)	0.003
MPO, ng/gm	1932 (756.5–5238.5)	1238.5 (663–2103.5)	0.013
NEO, nmol/L	5944 (3423–8323)	5357.5 (3894–7433)	0.726
Cytokines			
IL-1b, pg/mL	0.84 (0.68–0.99)	0.40 (0–0.77)	<0.001
IL-4, pg/mL	3.41 (2.0–3.8)	1.5 (0–3.0)	0.026
IL-6, pg/mL	0 (0–1.2)	1.68 (0.83–4.7)	0.001
IL-10, pg/mL	17.3 (12.8–27.6)	20.7 (17.2–29.1)	0.007
IL-12, pg/mL	1.05 (0.08–2.4)	0.80 (0.42–1.5)	0.113
TNF, pg/mL	1.49 (0–7.7)	11.09 (8.8–15.3)	<0.001
IFN-γ, pg/mL	0 (0–2.5)	0 (0–0)	0.003
Systemic inflammatory marker			
CRP, mg/L	1.18 (0.58–2.2)	0.52 (0.15–2.3)	0.371

Abbreviations: AAT: alpha-1 antitrypsin; CRP: C-reactive protein; FGF: fibroblast growth factor; IFN-γ: gamma interferon; GLP: glucagon-like peptide; ILs: interleukins; IGF: insulin-like growth factor; MPO: myeloperoxidase; NEO: neopterin; OXM: oxyntomodulin; TNF-α: tumor necrosis factor-alpha. All values are medians; 25th–75th percentiles are shown in parentheses; *p*-values were determined by Wilcoxon matched-pair signed-rank test.

**Table 5 nutrients-11-01827-t005:** Association of child length and BMI change with adipokines and peptides at six months post-enrollment among stunted and control children.

Variables	Crude			Adjusted		
	Beta Co-Efficient	95% CI	*p*	Beta Co-Efficient	95% CI	*p*
**Length change (Δ length in cm) at 6 months post-enrolment**						
**^1^ Stunted children (*n* = 39)**						
Leptin, µg/L	0.27	−0.25, 0.78	0.298	0.28	−0.27, 0.85	0.306
Adiponectin, µg/mL	0.01	−0.07, 0.10	0.741	0.02	−0.08, 0.11	0.746
Leptin–adiponectin ratio	1.51	−5.84, 8.88	0.679	1.77	−7.07, 10.62	0.684
IGF-1, ng/mL	0.01	−0.02, 0.04	0.483	0.02	−0.01, 0.05	0.200
IFN-γ, pg/mL	0.10	−0.33, 0.55	0.620	0.008	−0.53, 0.55	0.974
IL-6, pg/mL	−0.033	−0.11, 0.04	0.422	−0.017	−0.12, 0.08	0.730
**^2^ Control children (*n* = 24)**						
Leptin, µg/L	0.29	−0.18, 0.75	0.211	0.23	−0.25, 0.71	0.327
Adiponectin, µg/mL	−0.10	−0.19, −0.009	0.032	−0.087	−0.18, 0.008	0.073
Leptin–adiponectin ratio	4.2	0.65, 7.7	0.023	3.7	−0.20, 7.5	0.062
IGF-1, ng/mL	0.001	−0.02, 0.02	0.880	0.003	−0.02, 0.02	0.728
IFN-γ, pg/mL	−0.02	−0.07, 0.03	0.438	−0.02	−0.07, 0.02	0.383
IL-6, pg/mL	−0.04	−0.274, 0.193	0.722	−0.066	−0.30, 0.17	0.571
**BMI kg/m^2^ change (Δ BMI kg/m^2^) at 6 months from enrolment**						
**^1^ Stunted children (*n* = 39)**						
Leptin, µg/L	0.04	−0.21, 0.30	0.740	−0.004	−0.29, 0.27	0.974
Adiponectin, µg/mL	0.01	−0.07, 0.10	0.741	−0.04	−0.09, 0.0006	0.053
Leptin–adiponectin ratio	3.8	0.50, 7.2	0.025	3.4	−0.80, 7.6	0.109
IGF-1, ng/mL	−0.01	−0.03, 0.002	0.080	−0.02	−0.03, 0.000	0.050
IFN-γ, pg/mL	−0.006	−0.22, 0.21	0.950	0.05	−0.22, 0.32	0.724
IL-6, pg/mL	0.02	−0.012, 0.067	0.166	0.04	−0.007,0.09	0.094
**^2^ Control children (*n* = 24)**						
Leptin, µg/L	0.04	−0.27, 0.35	0.779	0.01	−0.25 to 0.27	0.933
Adiponectin, µg/mL	−0.10	−0.19, −0.009	0.105	−0.08	−0.18, 0.008	0.042
Leptin–adiponectin ratio	−0.33	−2.9, 2.2	0.793	−0.92	−3.2, 1.3	0.403
IGF-1, ng/mL	0.003	−0.008, 0.015	0.567	0.007	−0.002, 0.02	0.131
IFN-γ, pg/mL	−0.005	−0.04, 0.03	0.705	0.004	−0.02, 0.03	0.718
IL-6, pg/mL	−0.02	−0.17, 0.12	0.760	0.02	−0.11, 0.15	0.757

Abbreviations: CI: confidence interval; IFN-γ: gamma interferon; IGF: insulin-like growth factor; IL: interleukin. ^1^ Model for stunted children was adjusted for child age, child sex, food intake, and child cognitive development. ^2^ Model for control children was adjusted for child age and child sex.

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
