# Peer review of "Serum Adipokines, Growth Factors, and Cytokines Are Independently Associated with Stunting in Bangladeshi Children"

_nutrients, 2019, doi:10.3390/nu11081827_

Round 1

Reviewer 1 Report

The Abstract is incomplete as it does not mention that a combined nutritional and psychosocial support was applied to the intervention group. Moreover, the abstract should also include the information that all children were clinically screened over the study period.

The authors applied an incomplete study design. Formally, they lack the “stunted, no intervention” group and the “non-stunted intervention” group of children. I understand that, for ethical reasons, there cannot be a “stunted, no intervention” group. However, the authors need to point that out in the Material and Methods section and need to discuss the logical implication of the incomplete study design in the Discussion, particularly with respect to the comparability of the results. The biggest problem is that, with the current study design, it cannot be decided whether, in the intervention group, the observed changes in the biomarkers occur as an effect of the intervention or are a naturally occurring effect based on the aging process of the children. While the control group is subjected to the time effect only, the intervention group is subjected to time x intervention (FS+PS). This applies particularly to the statistical tests described in l. 193-202 and to Table 4 where the controls and the initially stunted children were compared AFTER the intervention. The latter comparison applies to two groups that were exposed to two different exposures each and it is impossible to say whether the differences are either due to the aging x intervention or aging x (stunted/not stunted).

The Discussion builds to a large part on results that are in contrast to what is reported in the Result section. For example, the authors state in the Discussion that “Levels of CRP and most of the pro-inflammatory cytokines (IL-6, IL-12, and TNF-α,) were lower among stunted children than controls”. In the Results, the associated p values (Table 4) are all above 0.05. Hence, a considerable part of the Discussion lacks a solid basis.

The results of the study might be biased by the higher drop-out rate of children from the control group. This needs to be discussed.

l.30-31. As the children were grouped due to their z-scores, it seems redundant to indicate that a significant difference existed between the stunted and the control group. I suggest to remove this sentence.

l. 60 Remove “the” at the end of the line.

l. 61 I suggest to replace “adipose tissue or fat cells” by “adipocytes”.

l. 69 The sentence “In addition, leptin mediates growth by regulating the energy levels by suppression of appetite and stimulation of energy expenditure” is not precise – how should reduced appetite stimulate growth? Moreover, energy expenditure can ONLY be increased by increasing physical activity as you simply cannot just “waste” energy away by activating any kind of futile cycles. So, do the authors claim that increased leptin levels increase physical activity? I doubt that scientific evidence exists for that.

l. 72 It is not true that “Ghrelin is the only hormone that stimulates appetite” – think just of orexin or neuropeptide Y.

l. 72-74 The claim that “… this pathway is disrupted during infection and inflammation, conditions that are particularly common in children living in low-resource settings” is not substantiated by a reference.

l. 77 Replace “gamma interferon” by ”Interferon gamma”.

l. 79-81 I think that the authors should be more precise with the statement “ During infection, elevated levels of cytokines lead to increased blood leptin concentrations, diminished appetite [16], and antagonize growth”. Cytokines with a pro-inflammatory potential such as TNF-alpha may have this effect, but this may not hold true for cytokines suppressing inflammatory processes such as IL-10.

l. 85 Replace “heightened” with ”increased”.

l. 135-139 The authors should indicate clearly if this treatment was applied to all children or just the intervention group.

l. 160 Replace “interleukin’s” with “interleukins” (no genitive) and use “ILs” instead of “IL’s” as abbreviation.

l. 166-167 Were the kits for the analysis of myeloperoxidase, neopterin and alpha-1 antitrypsin also ELISAs? If so, mention of indicate the applied test principle and indicate how the stool samples were pre-treated before being applied to the assays.

l. 173 Indicate that “icddr,b” stands for “ International Centre for Diarrhoeal Disease Research, Bangladesh”.

l. 182 Change “Qualitative variables” to ”Counts”.

l. 188 Explain the abbreviations of the information criteria (AIC and BIC).

l. 230 and following: The authors should explain how they calculated the AOR values. Odds ratios typically refer to the ratios of counts due to a categorization. Did the authors fix for each of the markers a threshold value (e.g. being below or above a certain cytokine concentration) and calculated the OR whether this threshold concentration was exceeded in the control or intervention group? Moreover, “leptin: adiponectin” should be called ”leptin-adiponectin ratio”.

In Tables 2 and 3, IL-1B should be written as IL-1beta or IL-1b.

l. 300 Table 2 should be omitted – there is hardly anything to read from this.

l. 316 Should say “Tables”

l. 334-336 when the authors state that “This situation reverses with the improvement in child nutritional status. At the end of 6 months of intervention, there were higher levels of leptin, lower levels of adiponectin and an increase in leptin: adiponectin found among the stunted children.”, how do they know that this effect in stunted children is based on the nutritional intervention and does not occur as a consequence of the increased age of the children? The authors lack the corresponding control group (stunted, no intervention) in order to be able to make this statement.

l. 351-352 When the authors write that “Sub-optimal nutrition and chronic inflammation mediated by pro-inflammatory cytokines contribute to the underlying etiology of growth retardation” I think that they miss out the fact that undernutrition may weaken the body’s defenses against (gastrointestinal) infections, which in turn trigger the inflammation along with markers of inflammation. I don’t think that the picture is complete without including the wasting effect of the infection itself.

l. 352 and following: The statement “Leptin stimulates the proliferation of cells of the immune system, such as tumor necrosis factor-α, interleukin (IL)-2, or IL-6 or IL-12” is nonsense as cytokines are no cells. Moreover, leptin is an indicator of the saturation of fat content in adipocytes and not primarily a hormone intended to regulate inflammation, which is primarily controlled by MAMPs and comparable primary receptors recognizing microbial structures. The authors did not measure any significant differences either in IL-12 or leptin between stunted and non-stunted children so their discussion is bare of any results measured. Throughout the Discussion, the authors should limit their considerations only to the significant findings.

L. 354 + 357 and following: The authors write that “The raised levels of IL-12 and leptin observed among the stunted study children…” and “Levels of CRP and most of the pro-inflammatory cytokines (IL-6, IL-12, and TNF-α,) were lower among stunted children than controls.” This statement stands in contrast to the results listed in Table 4, where none of the associated p values is below 0.05, so it’s unclear why the authors claim to have measured significant differences although the reported results show the opposite.

l. 377-380 The first two sentences of this paragraph are redundant. Remove them.

Remove the word “Abbreviation” that occurs in all references in the list of references.

Author Response

Response to Reviewer 1 Comments

The Abstract is incomplete as it does not mention that a combined nutritional and psychosocial support was applied to the intervention group. Moreover, the abstract should also include the information that all children were clinically screened over the study period.

Response: Thank you for your comment. We agree with you and revised the abstract accordingly.

The authors applied an incomplete study design. Formally, they lack the “stunted, no intervention” group and the “non-stunted intervention” group of children. I understand that, for ethical reasons, there cannot be a “stunted, no intervention” group. However, the authors need to point that out in the Material and Methods section and need to discuss the logical implication of the incomplete study design in the Discussion, particularly with respect to the comparability of the results. The biggest problem is that, with the current study design, it cannot be decided whether, in the intervention group, the observed changes in the biomarkers occur as an effect of the intervention or are a naturally occurring effect based on the aging process of the children. While the control group is subjected to the time effect only, the intervention group is subjected to time x intervention (FS+PS). This applies particularly to the statistical tests described in l. 193-202 and to Table 4 where the controls and the initially stunted children were compared AFTER the intervention. The latter comparison applies to two groups that were exposed to two different exposures each and it is impossible to say whether the differences are either due to the aging x intervention or aging x (stunted/not stunted).

Response: We agree with your comments and revised the manuscript. In addition, we also reanalyzed and revised table 5 showing the time-group interaction to find out the effect of intervention among the groups. We also revised our result and discussion section according to the study findings.

The Discussion builds to a large part on results that are in contrast to what is reported in the Result section. For example, the authors state in the Discussion that “Levels of CRP and most of the pro-inflammatory cytokines (IL-6, IL-12, and TNF-α,) were lower among stunted children than controls”. In the Results, the associated p values (Table 4) are all above 0.05. Hence, a considerable part of the Discussion lacks a solid basis.

Response: We revised the discussion section accordingly.

The results of the study might be biased by the higher drop-out rate of children from the control group. This needs to be discussed.

Response: We have discussed the higher drop out rate in the study limitation section.

l.30-31. As the children were grouped due to their z-scores, it seems redundant to indicate that a significant difference existed between the stunted and the control group. I suggest to remove this sentence.

Response: We removed the sentence.

l. 60 Remove “the” at the end of the line.

Response: We removed “the” at the end of the line.

l. 61 I suggest to replace “adipose tissue or fat cells” by “adipocytes”.

Response: We replaced “adipose tissue or fat cells” with “adipocytes”.

l. 69 The sentence “In addition, leptin mediates growth by regulating the energy levels by suppression of appetite and stimulation of energy expenditure” is not precise – how should reduced appetite stimulate growth? Moreover, energy expenditure can ONLY be increased by increasing physical activity as you simply cannot just “waste” energy away by activating any kind of futile cycles. So, do the authors claim that increased leptin levels increase physical activity? I doubt that scientific evidence exists for that.

Response: We revised the sentence.

l. 72 It is not true that “Ghrelin is the only hormone that stimulates appetite” – think just of orexin or neuropeptide Y.

Response: We revised the statement.

l. 72-74 The claim that “… this pathway is disrupted during infection and inflammation, conditions that are particularly common in children living in low-resource settings” is not substantiated by a reference.

Response: We added a supportive reference to the statement.

l. 77 Replace “gamma interferon” by “Interferon gamma”.

Response: We replaced “gamma interferon” with “Interferon gamma”.

l. 79-81 I think that the authors should be more precise with the statement “ During infection, elevated levels of cytokines lead to increased blood leptin concentrations, diminished appetite [16], and antagonize growth”. Cytokines with a pro-inflammatory potential such as TNF-alpha may have this effect, but this may not hold true for cytokines suppressing inflammatory processes such as IL-10.

Response: We agree with your comment and we revised the statement.

l. 85 Replace “heightened” with “increased”.

Response: We replaced “heightened” with “increased”.

l. 135-139 The authors should indicate clearly if this treatment was applied to all children or just the intervention group.

Response: The treatment “routine clinical care” was applied to all children.  

l. 160 Replace “interleukin’s” with “interleukins” (no genitive) and use “ILs” instead of “IL’s” as abbreviation.

Response: We replaced “interleukin’s” with “interleukins” and used “ILs” instead of “IL’s” as an abbreviation.

l. 166-167 Were the kits for the analysis of myeloperoxidase, neopterin and alpha-1 antitrypsin also ELISAs? If so, mention of indicate the applied test principle and indicate how the stool samples were pre-treated before being applied to the assays.

Response: We have added the suggested information.

l. 173 Indicate that “icddr,b” stands for “ International Centre for Diarrhoeal Disease Research, Bangladesh”.

Response: It is now recommended by the organization to use the abbreviation icddr,b as its name. We are directing a quote from icddr,b’s branding guideline:

“We used to be known as the International Centre for Diarrhoeal Disease Research, Bangladesh, but given the scope of our work is so much greater, we now use the abbreviation of our name – icddr,b.”

l. 182 Change “Qualitative variables” to “Counts”.

Response: We changed it.

l. 188 Explain the abbreviations of the information criteria (AIC and BIC).

Response: We explained the abbreviations of the information criteria (AIC and BIC).

l. 230 and following: The authors should explain how they calculated the AOR values. Odds ratios typically refer to the ratios of counts due to a categorization. Did the authors fix for each of the markers a threshold value (e.g. being below or above a certain cytokine concentration) and calculated the OR whether this threshold concentration was exceeded in the control or intervention group? Moreover, “leptin: adiponectin” should be called “leptin-adiponectin ratio”.

In Tables 2 and 3, IL-1B should be written as IL-1beta or IL-1b.

Response:  We did not fix for a threshold value for each of the markers to calculate the OR. We calculated the adjusted OR (AOR) by adjusting the statistical model (logistic model) for potential confounders; which were child age, sex, weight-for-age Z-score and hemoglobin status in this case.

We changed “leptin: adiponectin” to “leptin-adiponectin ratio” throughout the manuscript.

We also changed “IL-1B” to IL-1b in Tables 2 and 3.

l. 300 Table 2 should be omitted – there is hardly anything to read from this.

Response: Thank you for your suggestion. We assume that you meant Figure 2. We have omitted Figure 2 from the manuscript.

l. 316 Should say “Tables”

Response: We revised it.

l. 334-336 when the authors state that “This situation reverses with the improvement in child nutritional status. At the end of 6 months of intervention, there were higher levels of leptin, lower levels of adiponectin and an increase in leptin: adiponectin found among the stunted children.”, how do they know that this effect in stunted children is based on the nutritional intervention and does not occur as a consequence of the increased age of the children? The authors lack the corresponding control group (stunted, no intervention) in order to be able to make this statement.

Response: We agree that we had no corresponding control group (stunted, no intervention) in this study, which is a limitation of this study and we have mentioned about it in study limitations under the discussion section. We also agree with you regarding the increased child age with adipokine changes. We did further analysis by adjusting for time-group interaction and revised the result and discussion accordingly.

l. 351-352 When the authors write that “Sub-optimal nutrition and chronic inflammation mediated by pro-inflammatory cytokines contribute to the underlying etiology of growth retardation” I think that they miss out the fact that undernutrition may weaken the body’s defenses against (gastrointestinal) infections, which in turn trigger the inflammation along with markers of inflammation. I don’t think that the picture is complete without including the wasting effect of the infection itself.

Response: We have added the suggested information.

l. 352 and following: The statement “Leptin stimulates the proliferation of cells of the immune system, such as tumor necrosis factor-α, interleukin (IL)-2, or IL-6 or IL-12” is nonsense as cytokines are no cells. Moreover, leptin is an indicator of the saturation of fat content in adipocytes and not primarily a hormone intended to regulate inflammation, which is primarily controlled by MAMPs and comparable primary receptors recognizing microbial structures. The authors did not measure any significant differences either in IL-12 or leptin between stunted and non-stunted children so their discussion is bare of any results measured. Throughout the Discussion, the authors should limit their considerations only to the significant findings.

Response: We agree with your comment and revised the statement.

L. 354 + 357 and following: The authors write that “The raised levels of IL-12 and leptin observed among the stunted study children…” and “Levels of CRP and most of the pro-inflammatory cytokines (IL-6, IL-12, and TNF-α,) were lower among stunted children than controls.” This statement stands in contrast to the results listed in Table 4, where none of the associated p values is below 0.05, so it’s unclear why the authors claim to have measured significant differences although the reported results show the opposite.

Response: We agree with you and revised the statement in the discussion section.

l. 377-380 The first two sentences of this paragraph are redundant. Remove them.

Response: We removed the sentences.

Remove the word “Abbreviation” that occurs in all references in the list of references.

Response: Thank you for your keen observation. We revised the reference section.

Reviewer 2 Report

Comments to the Authors,

The topic of this manuscript is very important in child health and public health nutrition in Bangladesh and other countries. My comments are as follows.

Major comments:

Although the aim of this study is described at the end of introduction section, readers will want to know the implication of measurement of adipokines, growth factors, and cytokines in stunted children.

Main findings of the study was clearly described in results section, being discussed well in discussion section.

Regarding to Figure 2 (showing the changes in fasting serum leptin concentration by child length), its interpretation is not clear for me. Is the changes significantly different between stunted children and control children?

Minor comments:

 Please follow instructions for authors on research ethics: The study involves human and includes interventional methods. Therefore, as a minimum, a statement including the statement of the project identification code, date of approval and name of the ethics committee or institutional review board should be cited in the methods section.

Figure 1 is difficult to read.

Typo error, page7 line 247 and page 8 line 262 ; (table -Table)

Author Response

Response to Reviewer 2 Comments

Major comments:

Although the aim of this study is described at the end of introduction section, readers will want to know the implication of measurement of adipokines, growth factors, and cytokines in stunted children.

Response: Thank you for the query. We now have added the implication of measurement of adipokines, growth factors, and cytokines in stunted children at the end of the introduction section.

Main findings of the study was clearly described in results section, being discussed well in discussion section.

Response: Thank you for the comment. We appreciate your expertise.

Regarding to Figure 2 (showing the changes in fasting serum leptin concentration by child length), its interpretation is not clear for me. Is the changes significantly different between stunted children and control children?

Response: The length change was not significantly different between stunted children and control children but shows a slight increase over time. We have omitted Figure 2 as per reviewer 1’s suggestion.

Minor comments:

Please follow instructions for authors on research ethics: The study involves human and includes interventional methods. Therefore, as a minimum, a statement including the statement of the project identification code, date of approval and name of the ethics committee or institutional review board should be cited in the methods section.

Response: We also agree with your concern. We have added the suggested information under the ethical consideration part of the methods section.

Figure 1 is difficult to read.

Response: We have revised and uploaded a better figure now.

Typo error, page7 line 247 and page 8 line 262 ; (table -Table)

Response: We have corrected the “table” to “Table”.
